# The Effects of Workload Excess on Quality of Work Life in Third-Level Healthcare Workers: A Structural Equation Modeling Perspective

**DOI:** 10.3390/healthcare12060651

**Published:** 2024-03-14

**Authors:** Mehmet Koca, Serdar Deniz, Feyza İnceoğlu, Ali Kılıç

**Affiliations:** 1Department of Health Management, Malatya Turgut Özal University, Malatya City 44210, Turkey; 2Department of Public Health, Malatya Turgut Özal University, Malatya City 44210, Turkey; sedar.deniz@ozal.edu.tr; 3Department of Biostatistics, Malatya Turgut Özal University, Malatya City 44210, Turkey; feyza.inceoglu@ozal.edu.tr; 4Rectorate Department, Malatya İnönü University, Malatya City 44280, Turkey; ali.kilic@inonu.edu.tr

**Keywords:** hospital, healthcare worker, workload, quality of work life

## Abstract

Examining the workload and working conditions of healthcare workers is crucial regarding the quality of the work life of employees and the quality of services provided. This study aims to determine the effects of the perceptions of workload on the quality of work life of health personnel working in two different tertiary hospitals affiliated with the Ministry of Health and the Higher Education Institution in Türkiye with structural equation modeling. This study was conducted in two hospitals in Malatya province: İnönü University Turgut Özal Medical Center affiliated with the Higher Education Institution and Malatya Training and Research Hospital affiliated with the Ministry of Health. The mean score of the Individual Workload Perception Scale was 106.18 ± 16.5, and the mean score of the Work-related Quality of Life Scale was 66.39 ± 15.33. Total workload scores had a statistically significant positive effect on total quality of work life scores (β1 = 0.98; *p* = 0.001). It was concluded that there was a statistically significant relationship between workload and quality of work life and age, unit of employment, working time in the profession, and hospital type.

## 1. Introduction

In general terms, workload is the volume of work per employee within an organization. When explaining workload, the amount of production per person, working hours, and the mental processes of the job to produce this production are also considered [1]. When workload is evaluated individually based on employees, it indicates the energy and time required for the work to be carried out; when evaluated from the organization’s perspective, it refers to efficiency and performance. The concept of workload, which has become one of the most fundamental issues of business life, is defined in other words as the amount of work that an employee with certain knowledge and experience must carry out in a specified period [2].

How working individuals objectively perceive their workload, regardless of the workload level, is called workload perception. When the individual thinks that the resources they have to carry out the job are sufficient for the job requirements, they perceive the workload as normal; when the individual thinks that the resources they have are insufficient and the job requirements are more than the available resources, they perceive the workload as high. When the individual thinks that the resources they have are too much for their current job, they may perceive their workload as low [3].

These perceptions in individuals, especially the perception of excessive workload, cause both mental and physical negativities in the working personnel and can affect the performance, efficiency, and even health of the personnel. It has become a necessity to have a highly motivated and qualified workforce to provide quality healthcare services, which is a labor-intensive sector. Perception of workload in the healthcare sector is closely related to the quality of patient care and affects burnout, job satisfaction, and intentions to leave [4,5]. Thus, workload is among the determinants of quality of work life [6].

Quality of work life is conceptually defined as “humanization of work” in Germany, “improvement of working conditions” in France, and “protection of workers” in Eastern European countries [7]. Quality of work life is related to the improvement of conditions for employees within the organization. Organizations with a reward system, where employees are given the opportunity to develop themselves regarding career, and where the working environment is adjusted to ergonomic conditions, can provide their employees with a high quality of work life [8]. Examining the workload and working conditions of healthcare professionals is crucial for the quality of life of employees and the quality of services provided [9].

In a study conducted on 355 healthcare personnel working in a private chain healthcare institution operating in various cities of Türkiye and Istanbul, it was determined that there was a negative significant relationship between workload and quality of work life, and a positive significant relationship between perceived workload and intention to quit [10]. In another study conducted on 324 healthcare professionals working in a public hospital in Türkiye, the mediating role of excess workload was tested, and in the established model, it was determined that excess workload negatively affected job satisfaction and employee performance [11].

In a study conducted at Başkent University Ankara Hospital to determine the effect of the workload of nurses working in surgical clinics on patient safety, it was determined that workload affected patient care and patient safety. It was concluded that increasing workload reduces the time devoted to the patient and increases the risk of errors as a result of fatigue and carelessness [12]. Sarıdiken and Çınar (2021) stated that excessive workload in the healthcare sector negatively affects the performance of employees, the quality of patient care, and patient safety. They also stated that measuring workload perception levels regarding the working environment is important to increase awareness [13].

In today’s working environment, the most important source of production is ensuring the quality of the work life of employees, increasing the organizational commitment of employees, and ensuring that employees continue to work for a long time. Trained human resources are extremely important in ensuring the competitive advantage of organizations against other organizations. Therefore, studies should be conducted to determine the workload perception and quality of work life of healthcare professionals working in institutions serving human health; management style is very important in terms of guiding the design of the work environment, organizational relations, and workload distribution and increasing the quality of the service provided.

This study aims to determine the effects of the perceptions of workload on the quality of work life of health personnel working in two different tertiary hospitals affiliated with the Ministry of Health and the Higher Education Institution with structural equation modeling.

## 2. Materials and Methods

### 2.1. Type and Hypotheses of This Research

In the design of our study, the relational survey model was used to examine the interactions between sets of variables [14]. The relationships between Individual Workload Perception and Work-related Quality of Life were examined.

The hypotheses of this study are stated below:

**H_1_:** *Workload and Work-related Quality of Life scores are affected by demographic variables*.

**H_2_:** *Workload and Work-related Quality of Life scores are affected by the work unit*.

**H_3_:** *Workload and Work-related Quality of Life scores are affected by the length of employment in the unit*.

**H_4_:** *Workload and Work-related Quality of Life scores are affected by the length of time working in the profession*.

**H_5_:** *Workload and Work-related Quality of Life scores are affected by hospital type*.

**H_6_:** *The effect of workload and Work-related Quality of Life scores is statistically significant*.

### 2.2. Place and Time This Research Was Conducted

This study was conducted in two of the largest hospitals in Malatya province. One of these hospitals is İnönü University Turgut Özal Medical Center, affiliated with the Higher Education Institution, and the other is Malatya Training and Research Hospital (MTRH), affiliated with the Ministry of Health. Turgut Özal Medical Center (TÖMC) has a bed capacity of 1585 and Malatya Training and Research Hospital has a bed capacity of 1055. The study data were collected through online methods and face-to-face between 1 July and 15 August 2023. Before starting this study, a voluntary consent form was presented to the participants, and this study was completed with the participants who agreed to sign the form.

### 2.3. Sample Selection and Number of Samples

It is emphasized in the literature that the number of samples should be over 200 in structural equation modeling analyses [14,15]. The snowball sampling method was used to select the participants in this study and 345 people participated. Seventeen of the participants filled out the survey form incompletely. Incompletely filled survey forms were not included in the study data, and this study was completed with the data obtained from 328 participants.

### 2.4. Data Collection Tools

#### 2.4.1. Personal Information Form

This is a 9-question socio-demographic survey form created by the authors to determine the participants’ age, gender, marital status, education, profession, hospital worked, years of employment, and length of time working in the unit.

#### 2.4.2. Individual Workload Perception Scale (IWPS)

The Individual Workload Perception Scale was developed by Cox (2003), and its validity and reliability analysis was conducted in 2006 [16]. The scale was adapted to Turkish by Saygılı (2008). A Turkish validity and reliability study of the scale was conducted by Sarıdiken and Çınar (2021). The “Individual Workload Perception Scale” consists of 31 items and five dimensions. These subscales are Manager Support, Peer Support, Unit Support, Workload, and Intent to Stay. The scale was created with a 5-point Likert-type scale. The numberings correspond to “1—Strongly Disagree, 2—Disagree, 3—Undecided, 4—Agree, 5—Strongly Agree” [13,17].

#### 2.4.3. The Work-Related Quality of Life (WRQoL) Scale

The Work-related Quality of Life (WRQoL) Scale was developed by Van Laar et al. (2007). Its adaptation into Turkish and a validity and reliability study were conducted by Akar and Üstüner (2017). The “Work-related Quality of Life Scale” consists of 23 items. There are six dimensions in the scale. These subscales are “Job and Career Satisfaction, General Well-Being, Control at Work, Stress at Work, Working Conditions, and Home-Work Interface”. The scale is a 5-point Likert-type scale. The scoring of the scale is 1 = Disagree, 2 = Agree slightly, 3 = Agree moderately, 4 = Agree mostly, 5 = Agree completely [18,19].

### 2.5. Statistical Analysis

The analysis of the data included in this research was carried out with the SPSS (Statistical Program in Social Sciences) 26.0 program. Whether the data included in the study complied with normal distribution was checked with the Kolmogorov–Smirnov Test [20]. For comparison tests, the significance level (*p*) was taken as 0.05. Since normal distribution was achieved in the variables (*p* > 0.05), the analysis was continued with parametric test methods. Descriptive values of the variables are given as numbers, percentages, mean, and standard deviation. Cronbach’s α coefficient was used to determine the reliability of the scales. A Structural Equation Model (SEM) was established and analyzed with the AMOS 24 package program. Multivariate analysis controls were provided for SEM. For a multivariate normal distribution, the coefficient for the “Observations farthest from the centroid (Mahalanobis Distance)” endpoint was 6.217 with AMOS. A coefficient less than 8 indicates a multivariate normal distribution [21].

The SEM was established using the variables observed in the study, the relationships between the variables were examined, and the model Goodness of Fit coefficients were evaluated. To explore the relationships between scale scores and variables, an independent *t*-test was used in two-group data, and an ANOVA test was used in multiple groups. After the ANOVA test, Duncan’s test was applied for the groups with homogeneity of variance, and the Tamhane T2 test was applied for the groups without homogeneity.

### 2.6. Ethics

Ethical approval for this study was obtained from the non-interventional ethics committee of Malatya Turgut Özal University with a letter dated 23 May 2023 and decision numbered E-30785963-020-158266. This research was conducted in accordance with the Principles of the Declaration of Helsinki.

## 3. Results

Demographic information about the participants included in the study is given in the table below.

This study was conducted with n = 328 participants. Of these participants, 54.3% (n = 178) work at Malatya Training and Research Hospital, and 45.7% work at İnönü University Turgut Özal Medical Center. Again, 74.4% (n = 244) of the participants were female, 22.9% (n = 75) were between the ages of 20 and 25, 59.8% (n = 196) were married, and 78.0% (n = 256) were undergraduate graduates. It was determined that 72.6% (n = 238) of the participants were nurses, 37.5% (n = 123) had been working in the profession for five years or less, 27.4% worked in emergency and internal branch services, and 59.5% (n = 195) had been working in the unit for fewer than five years (Table 1).

### 3.1. Descriptive Statistics of Scale Scores

The Individual Workload Perception Scale and its subscales of Manager Support, Peer Support, Unit Support, Workload, and Intent to Stay and the Work-related Quality of Life Scale and its subscales of Work Career Satisfaction, General Well-Being, Working Conditions, Stress at Work, and Home–Work Interface descriptive statistics and reliability coefficient Cronbach’s α values are given in Table 2.

The fact that the Cronbach’s α values of the scales used were higher than 0.70 showed that the reliability was high [22].

The mean score of the IWPS was calculated as 106.18 ± 16.5 and Cronbach’s α value was 0.819, and the mean score of the WRQoL was calculated as 66.39 ± 15.33 and Cronbach’s α value was 0.928 (Table 2).

### 3.2. Comparisons According to Scale Scores

A comparison of the Individual Workload Perception Scale and its subscales of Manager Support, Peer Support, Unit Support, Workload, and Intent to Stay and the Work-related Quality of Life Scale and its subscales of Job and Career Satisfaction, General Well-Being, Working Conditions, Stress at Work, and Home–Work Interface scores was conducted according to work unit, years of employment, years of working in the unit, and hospital type.

There was a statistically significant difference in the scores of the IWPS and its subscales, Manager Support, Peer Support, Unit Support, Workload, and Intent to Stay, according to age groups (*p* < 0.05). There was a statistically significant difference in unit support scores according to gender (*p* < 0.05). There was a statistically significant difference in the scores of the IWPS and its subscales, Manager Support, Unit Support, Workload, and Intent to Stay, according to educational status (*p* < 0.05) (Table 3).

There was a statistically significant difference in the scores of the IWPS and its subscales, Manager Support, Peer Support, Unit Support, Workload, and Intent to Stay, according to the work unit (*p* < 0.05). There was a statistically significant difference in Intent to Stay scores depending on the length of time working in the unit (*p* < 0.05).

There was a statistically significant difference in the scores of the IWPS and its subscales, Manager Support, Unit Support, Workload, and Intention to Maintain the Current Job, according to the length of time in the profession (*p* < 0.05). There was a statistically significant difference in the scores of the IWPS and its subscales, Peer Support, Unit Support, and Workload, depending on hospital type (*p* < 0.05) (Table 3).

There was a statistically significant difference in the scores of the WRQoL and its subscales Job and Career Satisfaction, General Well-Being, Control at Work, Working Conditions, Stress at Work, and Home–Work Interface according to age groups (*p* < 0.05). There was a statistically significant difference in Job and Career Satisfaction scores according to gender (*p* < 0.05). No statistically significant difference was detected in the scores of the WRQoL and its subscales Job and Career Satisfaction, General Well-Being, Control at Work, Working Conditions, Stress at Work, and Home–Work Interface according to educational status (*p* > 0.05) (Table 4).

There was a statistically significant difference in the scores of the WRQoL and its subscales Job and Career Satisfaction, General Well-Being, Control at Work, Working Conditions, Stress at Work, and Home–Work Interface depending on the work unit (*p* < 0.05). There was a statistically significant difference in the Working Conditions and Stress at Work scores, which are the subscales of the WRQoL, depending on the length of time working in the unit (*p* < 0.05). There was a statistically significant difference in the scores of the WRQoL and its subscales, Job and Career Satisfaction, Control at Work, Working Conditions, Stress at Work, and Home–Work Interface, depending on the duration of employment in the profession (*p* < 0.05) (Table 4).

There was a statistically significant difference in the WRQoL and its subscales Job and Career Satisfaction, General Well-Being, Stress at Work, and Home–Work Interface scores depending on hospital type (*p* < 0.05) (Table 4).

### 3.3. Structural Equation Model (SEM) Mediation Analysis

A model was established for SEM analysis and the relationships between Individual Workload Perception and the Work-related Quality of Life Scale were examined. The diagram below is given for the established model.

Individual Workload Perception Scale scores symbolize the independent variable. Work-related Quality of Life Scale scores symbolize the independent variable, and expressions e1–e12 symbolize the residual terms. The coefficients of the model are given in Table 5.

Since the Intention to Maintain the Current Job subscale score in the model was not statistically significant (*p* = 0.785), it was excluded from the model and further analysis. In the new model established, workload scores explained 96% of work life quality scores (R^2^ = 0.96, Table 5, Figure 1). In the modified model, total workload scores had a statistically significant positive effect on the total quality of work life scores (β1 = 0.98; *p* = 0.001). Only a 1-unit score increase in a person’s workload score will cause a 0.98-unit score increase in the work life quality scale score (β1 = 0.98, Figure 1, Table 5).

Fit indices were evaluated to test the significance of the established model. In the established model, the CMIN value was 123.313 χ^2^/df (Chi-Square Goodness of Fit; χ^2^, df; degree of freedom) 4.110. CFI (Comparative Fit Index) and IFI (Incremental Fit Index) values of 0.957, a NFI (Normed Fit Index) value of 0.944, and a GFI (Goodness of Fit Index) of 0.933 indicate that the model fit is very good (IFI > 0.90, NFI > 0.90, CFI > 0.90, GFI > 0.90). The fact that the RMSEA (Root Mean Square Error of Approximation) value used for the adequacy of the sample number is 0.074 (RMSEA < 0.80) shows that the sample size and the model used are sufficient [23].

## 4. Discussion

In the health sector, a branch of the service sector, workload closely concerns the quality of work life of the employees, their performance, and the quality of patient care. It is the key to organizations retaining their greatest capital, trained manpower. Studies have shown that workload positively affects the intention to leave the current job [24,25]. In a study conducted by Said and El-Shafei (2021) on nurses, 98.6% of nurses stated that workload was one of the most important reasons affecting their intention to leave work [26]. Another study showed that 80% of nurses who left the profession had intended to leave within the last year [27]. Some studies have shown a significant relationship between nurses’ turnover intention and demographic characteristics [28,29]. In this study, a significant relationship was found between the intention to continue in the current job and age, unit of employment, duration of employment in the unit, and total duration of employment. However, Çiftçioğlu et al. (2018) did not find a significant relationship between demographic variables and Intention to Maintain the Current Job [30].

Çiftçioğlu et al.’s (2018) study showed that the average score of the IWPS of healthcare professionals working in the hospital was 103.7 ± 22.07. In the subscales of the IWPS, the mean score for Manager Support was 25.94 ± 4.73, the mean score for Peer Support was 26.56 ± 5.18, the mean score for Unit Support was 19.25 ± 5.80, the mean score for Workload was 25.83 ± 4.49, and the mean score for Intent to Stay was 6.12 ± 1.87 [30].

Karacabay et al. (2020) found the mean score of surgical nurses’ workload perceptions to be 100.30 ± 12.79. They calculated the mean score of Manager Support, which is a subscale of the IWPS, as 25.32 ± 6.93, the mean score of Peer Support as 24.78 ± 4.94, the mean score of Unit Support as 22.50 ± 3.75, the mean score of Workload as 13.51 ± 3.04, and the mean score of Intent to Stay as 14.81 ± 2.67 [31].

In this study, the mean score of the IWPS was calculated as 106.18 ± 16.5. The mean score of Manager Support, which is a subscale of the IWPS, was 28.38 ± 7.13, the mean score of Peer Support was 29.68 ± 6.36, the mean score of Unit Support was 19.45 ± 4.26, the mean score of Workload was 19.63 ± 2.79, and the mean score of Intent to Stay was 9.02 ± 1.58. Using the same scale as this study, Çiftçioğlu et al. (2018) [30] and Karacabay et al. (2020) [31] reported that there were differences in the total score average of the IWPS and the scale and subscale score averages.

In our study, the score averages of Manager Support, Peer Support, and Intent to Stay were consistent with Çiftçioğlu et al.’s (2018) findings [30]; ours were higher than in their study. According to the study conducted by Karacabay et al. (2020) [31], the mean scores of Manager Support, Peer Support, and Workload were higher. It is thought that this difference arose from the sample sets, the fact that the hospitals where the studies were conducted were in different regions, and the differences between organizational cultures and business structure.

When the subscales of the IWPS were evaluated, the subscale with the highest score was the Peer Support subscale. When subscale weights were also considered during the evaluation, the same conclusion was reached. This result was found in the studies conducted by Cox et al. (2007) [16] on pediatric nurses and by Çiftçioğlu et al. (2018) [30] on hospital employees, and Karacabay et al.’s (2020) [31] study on surgical nurses was similar. Studies conducted in different years show that Peer Support continues. According to the results of our study, in the subscales of the IWPS, the highest score after Peer Support was the Manager Support subscale. This result is consistent with Çiftçioğlu et al. (2018) [30].

Among the subscales of the IWPS, the Unit Support subscale had the third-highest score. When this result is considered together with the Peer Support subscale, it is thought that the reason why the highest score was obtained from the Peer Support subscale, while the Unit Support subscale received fewer points, is due to the structure of the health sector and the fact that health professionals from different professional groups work in the unit as well as colleagues.

The Intent to Stay subscale of the IWPS was the lowest-scoring subscale of this scale. This result is consistent with Çiftçioğlu et al. (2018) [30]. The reason why the subscale of Intention to Stay received proportionally fewer points than the other subscales is that the healthcare workers in whom the research was conducted are permanent staff according to the Turkish Civil Servants Law. It is difficult to dismiss them without their consent, and also because the working conditions in the private sector are more difficult than the working conditions in public hospitals, which is thought to be due to lower wages.

According to the study results, a statistically significant difference was found between workload and age. While this result is similar to the results of a study conducted by Korkmazer (2021) on healthcare professionals in a public hospital in Muş province, it is different from the result of a study conducted by Çavuş (2023) on nurses working in surgical clinics at a university hospital [11,32].

According to the study results, no statistically significant difference was found between workload perception and gender. This result is parallel to other studies [30,31,32].

However, Göl (2019) concluded that there was a significant difference between workload scores according to gender in his study on nurses working in Pamukkale University Hospital [33].

According to the results of our study, no statistically significant difference was found between workload perception and educational status. This result is similar to the results of other studies [30,32]. However, Saygılı (2008) found a significant relationship between educational status and workload in her study on hospital employees [17].

In this study, while there was a significant difference between workload perception and working time in the profession, no significant difference was found between working time in the unit. Karacabey et al.’s (2020) study that is on surgical nurses is consistent [31]. However, Saygılı (2008) reported no significant relationship between working hours and workload in a study conducted on hospital employees [17].

IWPS scores show significant differences between hospitals. Regarding total workload scores, the score of employees at Malatya Training and Research Hospital was 109.14 ± 16.39, while the score of employees at İnönü University Turgut Özal Medical Center, another hospital where this study was conducted, was found to be 102.66 ± 15.99. In addition, in the subscales of the IWPS, except for the Intent to Stay subscale, it was determined that the scores of the employees of Malatya Training and Research Hospital were higher than the scores of the employees of İnönü University Turgut Özal Medical Center. This situation is important as it shows the situation of employees in two different organizational structures and climates. With this result, it is possible to say that according to the average workload scores, employees in the hospital affiliated with the Ministry of Health are offered a more positive organizational climate in terms of workload.

In this study, the average score of the WRQoL was 66.39 ± 15.33. In their study on nurses working in primary, secondary, and tertiary hospitals in Ghana, Poku et al. (2022) found the average score of the WRQoL to be 71.64 [34]. Similarly, McFadden et al. (2021), in their study on health professionals in the United Kingdom, found the average score of the WRQoL to be 78.15 [35]. There are differences between the results of their studies and our study. It is thought that this difference is due to the work structure, working conditions, and cultural reasons in the places where the studies were conducted.

A statistically significant difference was found between the age variable and quality of work life. In our study, it was determined that the group with the highest work life quality score was the group between the ages of 20 and 25. The significant difference we found between the age variable and quality of work life differs from the results of some studies in the literature [36,37,38]. In this study, no statistically significant difference was found between the gender variable and quality of work life. This result is consistent with some studies in the literature [38,39].

In this study, no statistically significant difference was found between the education variable and quality of work life. This result is similar to the results of some studies [37]. Torlak (2019) found a significant difference between educational level and quality of work life in a study conducted on nurses working in a private hospital in Istanbul [38]. Thakre et al. (2017) reported that the higher the educational level, the higher the quality of work life scores [40]. However, in the study conducted by Çatak and Bahçecik (2015), it was reported that as the educational level of nurses increased, the total score and subscale scores of the quality of nursing work life were negatively affected [36].

Torlak (2019) [38] conducted a study on nurses working in a private hospital in Istanbul and found a significant difference between the quality of work life according to the unit they work in and the duration of experience in the profession. In our study, similar to the result found by Torlak (2019), a significant difference was found according to the unit worked and the duration of working in the profession. According to the results of this study, it was determined that personnel who did not have much contact with patients and work in administrative units had the highest quality of work life scores. It was also determined that the group that was most satisfied with their working time in the profession was the group that worked for five years or fewer. The findings suggest that this situation depends on the morale and motivation in the first years of the profession.

In their study on nurses working in public and university hospitals in Trabzon, Tamer and Öztürk (2021) concluded that there was no significant difference between the working time in the unit and the quality of work life. However, Torlak (2019) reported a significant difference between working time in the unit and quality of work life in a study conducted on nurses working in a private hospital in Istanbul. While the result of our study is consistent with the result of the study conducted by Tamer and Öztürk (2021), the result of the study conducted by Torlak (2019) is different [38,41].

There is an inverse proportion between the scoring on the IWPS we used in this study and the workload. A high score on the IWPS should be interpreted as a low workload. In this regard, workload scores explain 96% of work life quality scores (R2 = 0.96, Table 5, Figure 1). In the modified model, workload total scores had a statistically significant positive effect on work life quality total scores (β1 = 0.98; *p* = 0.001). A 1-unit increase in a person’s workload score will cause a 0.98-unit increase in the work life quality scale score (β1 = 0.98, Figure 1, Table 5). Increasing workload causes the quality of work life to decrease. This result coincides with some studies in the literature [9,42].

In our study, the average IWPS of employees at Malatya Training and Research Hospital was 109.14 ± 16.39, and the average IWPS of employees at İnönü University Turgut Özal Medical Center was 102.66 ± 15.99 (Table 3). The average work life quality score of employees at Malatya Training and Research Hospital was 69.38 ± 16.18, and the average IWPS score of employees at İnönü University Turgut Özal Medical Center was 62.86 ± 13.47. This result is extremely meaningful, as a high score on the IWPS is interpreted as a low workload. According to this result, it was determined that the quality of work life of employees at Malatya Training and Research Hospital was higher than that of employees at İnönü University Turgut Özal Medical Center. This result suggests that, regarding workload perception and quality of work life, employees working in hospitals affiliated with the Ministry of Health work in a more suitable environment than those working in university hospitals.

According to the study results, in contrast to the hypothesis “H_1_: Workload and work-related quality of life scores are affected by demographic variables”, the workload and quality of work life are affected only by the age variable (Table 4 and Table 5). No significant difference was found between gender and educational status variables and workload and quality of work life. Therefore, the H_1_ hypothesis was partially accepted. Regarding “H_2_: Workload and work-related quality of life scores are affected by work unit”, according to the study results, the H_2_ hypothesis was accepted because there was a significant difference between the unit worked, workload perception and quality of work life (Table 4 and Table 5). Regarding “H_3_: Workload and work-related quality of life scores are affected by the length of employment in the unit”, according to the study results, the H_3_ hypothesis was rejected because there was no significant relationship between working time in the unit, workload perception, and quality of work life (Table 4 and Table 5). Regarding “H_4_: Workload and work-related quality of life scores are affected by the length of time working in the profession”, according to the study results, the H_4_ hypothesis was accepted because there was a significant relationship between the duration of work in the profession, workload perception, and quality of work life. Regarding “H_5_: Workload and work-related quality of life scores are affected by hospital type”, according to the study results, the H_5_ hypothesis was accepted since there was a significant difference between workload perception and quality of work life and the type of hospital worked (Table 4 and Table 5). Regarding “H_6_: The effect of workload and work-related quality of life scores is statistically significant”, the last hypothesis of our study, workload was found to affect the quality of work life, and the H_6_ hypothesis was accepted (Figure 1, Table 5).

## 5. Conclusions

Regarding the perception of workload and quality of work life, it was concluded that it was affected by age, unit of work, length of time working in the profession, and hospital type. However, regarding the perception of workload and quality of work life, it was determined that it was not affected by gender, educational level, and working time in the unit. According to the study results, there was a significant relationship between workload and quality of work life. Workload scores explained 96% of work life quality scores. A 1-unit decrease in workload caused a 0.98-unit increase in the Work-related Quality of Life (WRQoL) Scale. It has been concluded that increasing workload causes the quality of work life to decrease.

As in other service sectors, in the health sector, the quality of work life must be high for employees to be loyal and productive to the institution they work for and to make this sustainable. High quality of work life depends on workload. Given that there is a high correlation between workload and quality of work life, it should be taken into account that the subscales of employees’ workload perception depend on manager, colleague, and unit support, as well as the work environment, and it should be noted that employee-oriented policies should be followed in all areas. To increase the quality of work life, it is necessary to balance the workload distribution among employees and create a suitable organizational climate. For this, it can benefit all stakeholders, especially managers, to act together.

## Figures and Tables

**Figure 1 healthcare-12-00651-f001:**
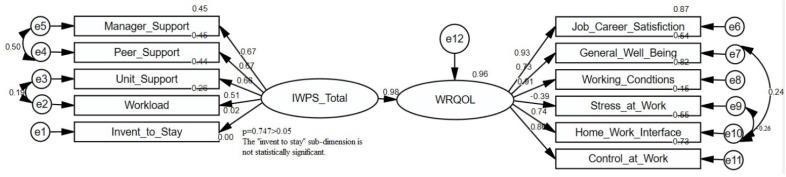
AMOS diagram of the relationship between individual workload and quality of work life.

**Table 1 healthcare-12-00651-t001:** Demographic information of participants (n = 328).

Variable	Group	Number	Percentage
Working Hospital	Malatya Training and Research Hospital	178	54.3
	Turgut Özal Medical Center	150	45.7
Gender	Female	244	74.4
	Male	84	25.6
Age Group	20–25 years old	75	22.9
	26–30 years old	63	19.2
	31–40 years old	66	20.1
	41–46 years old	64	19.5
	47 years and older	60	18.3
Marital Status	Married	196	59.8
	Single	123	37.5
	Separated/Widowed	9	2.7
Educational Status	High school	18	5.5
	Associate Degree	21	6.4
	Bachelor’s Degree	256	78.0
	Master’s Degree	33	10.1
Profession	Midwife	63	19.2
	Nurse	238	72.6
	Medical Officer	18	5.5
	Other	9	2.7
Working Time in the Profession	5 years or less	123	37.5
	6–10 years	24	7.3
	11–15 years	39	11.9
	16–20 years	50	15.2
	21–25 years	48	14.6
	26 years and over	44	13.4
Worked Unit	Emergency Service	90	27.4
	Operating Room Department	13	4.0
	Intensive Care Unit	39	11.9
	Polyclinic Unit	15	4.6
	Internal Branch Services	90	27.4
	Surgical Branch Services	70	21.3
	Other	11	3.4
Working Time in the Unit	Less than 5 years	195	59.5
	5–9 years	69	21.0
	10–14 years	37	11.3
	15–19 years	15	4.6
	20–25 years	6	1.8
	26–29 years	6	1.8

**Table 2 healthcare-12-00651-t002:** Descriptive statistics of scale scores.

Scale	Mean ± sd	Min–Max	Cronbach’s α
Manager Support	28.38 ± 7.13	8–40	0.702
Peer Support	29.68 ± 6.36	11–40	0.741
Unit Support	19.45 ± 4.26	9–30	0.711
Workload	19.63 ± 2.79	12–30	0.736
Intent to Stay	9.02 ± 1.58	6–15	0.786
IWPS Total	106.18 ± 16.5	57–145	0.819
Job and Career Satisfaction	18.45 ± 4.97	7–30	0.769
General Well-Being	17.51 ± 3.72	10–27	0.814
Working Conditions	8.15 ± 3.13	3–15	0.904
Control at Work	7.51 ± 3.28	3–15	0.729
Stress at Work	5.76 ± 2.17	2–10	0.769
Home–Work Interface	9.03 ± 3.2	3–15	0.794
WRQoL Total	66.39 ± 15.33	32–112	0.928

sd: standard deviation.

**Table 3 healthcare-12-00651-t003:** Comparison of Individual Workload Perception scale scores according to work unit, years of employment, years of working in the unit, and hospital type.

Variables	Groups	Manager Support	Peer Support	Unit Support	Workload	Intent to Stay	IWPS Total
		Mean ± sd	Mean ± sd	Mean ± sd	Mean ± sd	Mean ± sd	Mean ± sd
Age Group	20–25 age ^1^	31.6 ± 4.8	31.92 ± 5	21.32 ± 4.26	20.6 ± 3.55	9.44 ± 1.89	114.88 ± 15.72
	26–30 age ^2^	25.9 ± 7.41	27.43 ± 7.03	17 ± 3.9	18.62 ± 2.76	9.38 ± 1.6	98.33 ± 15.22
	31–40 age ^3^	28.23 ± 6.77	28.73 ± 6.8	19.09 ± 3.82	18.77 ± 1.94	8.86 ± 1.15	103.68 ± 15.6
	41–46 age ^4^	30.02 ± 6.45	31.75 ± 4.77	19.34 ± 3.7	20.23 ± 1.77	8.52 ± 1.47	109.86 ± 12.28
	47 age and above ^5^	25.4 ± 8.27	28.1 ± 6.78	20.2 ± 4.44	19.8 ± 2.9	8.85 ± 1.47	102.35 ± 18.11
Test ^a^ (F)		10.237	7.896	10.504	7.199	4.291	12.234
*p*		0.001 *	0.001 *	0.001 *	0.001 *	0.002 *	0.001 *
Difference		1–2, 3, 4, 5, 2–4, 3–5	1–2, 4, 2, 4, 5, 3–4, 4–5	1–3, 4, 5, 2–5, 3–5	1–3, 4, 5, 2–3, 4, 3–4, 4–5	1–2, 4, 3–5, 4–5	1–2, 5, 2–4, 3–4, 4–5
Gender	Female	28.63 ± 7.25	30.05 ± 6.4	19.75 ± 4.21	19.69 ± 2.76	8.96 ± 1.58	107.09 ± 16.83
	Male	27.68 ± 6.75	28.61 ± 6.16	18.57 ± 4.31	19.46 ± 2.9	9.21 ± 1.58	103.54 ± 15.31
Test ^b^ (*t*)		1.107	3.249	4.871	0.417	1.639	2.91
*p*		0.294	0.072	0.028	0.519	0.201	0.089
Education Status	High School	27.78 ± 8.82	30.33 ± 8.03	19.78 ± 3.95	20.83 ± 3.28	9.44 ± 1.65	108.17 ± 22.33
	Associate Degree	24 ± 6.46	28.14 ± 7.48	20.14 ± 4.97	20.86 ± 2.08	9.43 ± 1.54	102.57 ± 14.77
	Bachelor’s Degree	28.72 ± 6.99	29.76 ± 6.16	19.35 ± 4.28	19.47 ± 2.75	8.99 ± 1.59	106.29 ± 16.23
	Master’s Degree	28.91 ± 7	29.73 ± 6.39	19.64 ± 3.93	19.45 ± 2.98	8.82 ± 1.42	106.55 ± 16.5
Test ^a^ (F)		2.993	0.483	0.289	2.829	1.119	0.428
*p*		0.031 *	0.695	0.833	0.039 *	0.342	0.733
Difference		2–3			2–3		
Worked Unit	Emergency Service ^1^	30.27 ± 6.7	31.53 ± 5.34	21.33 ± 3.86	21.17 ± 2.8	9.1 ± 1.82	113.4 ± 16.3
	Operating Room ^2^	30 ± 5.23	30.92 ± 2.1	21 ± 1.87	19.15 ± 3.18	8.54 ± 1.45	109.62 ± 10.01
	Intensive Care Unit ^3^	24 ± 6.89	28.54 ± 7.06	17.69 ± 3.92	19 ± 2.06	9.62 ± 1.52	98.85 ± 16.88
	Polyclinic Unit ^4^	22.2 ± 7.53	23.8 ± 6.19	17.4 ± 4.07	19 ± 1.73	9.4 ± 1.68	91.8 ± 16.68
	Internal Branch Services ^5^	27.03 ± 7.86	28.3 ± 7.61	18.73 ± 5.24	18.93 ± 3.25	9.03 ± 1.52	102.03 ± 18.08
	Surgical Branch Services ^6^	30.53 ± 5.07	30.49 ± 5.15	18.67 ± 2.89	19.1 ± 1.75	8.51 ± 1.18	107.3 ± 10.44
	Other ^7^	32.45 ± 2.66	31.36 ± 2.94	22.09 ± 1.7	19.91 ± 2.7	9.55 ± 1.51	115.36 ± 4.99
Test ^a^ (F)		0.001 *	5.072	7.143	7.169	2.804	8.825
*p*		0.001 *	0.001 *	0.001 *	0.001 *	0.011	0.001 *
Difference		1–3, 4, 5, 2–4, 3–6, 7, 4–6, 7, 5–6	1–4, 2–6, 7, 3–6–7, 4–5, 5–6	1–2, 6, 7, 2–3, 6, 3–6, 4–5, 6	1–2, 3, 6, 7, 2–6, 3–4, 5, 6, 4–6	1–2, 6, 2–4, 5, 6, 3–4, 5, 4–5	1–4, 5, 6, 2–4, 5, 3–4, 5, 6–7
Working Time in Unit	<5 years ^1^	28.95 ± 6.85	29.91 ± 6.23	19.75 ± 4.49	19.62 ± 3.09	9.29 ± 1.61	107.52 ± 16.95
	5–9 years ^2^	27.17 ± 7.47	30.13 ± 5.75	19.04 ± 4.32	19.78 ± 2.85	8.87 ± 1.55	105 ± 14.74
	10–14 years ^3^	28.16 ± 7.84	28.81 ± 7.55	19.19 ± 3.56	19.43 ± 1.39	8.65 ± 1.18	104.24 ± 16.75
	15–19 years ^4^	25.6 ± 8.35	25.2 ± 8.19	16.6 ± 2.32	19.6 ± 1.68	7.6 ± 1.06	94.6 ± 17.83
	20–25 years ^5^	32.5 ± 2.74	33.5 ± 0.55	22.5 ± 1.64	19.5 ± 0.55	9.5 ± 1.64	117.5 ± 6.02
	26–29 years ^6^	28 ± 4.38	30 ± 2.19	20 ± 0	20 ± 2.19	7.5 ± 0.55	105.5 ± 9.31
Test ^a^ (F)		1.528	2.220	2.378	0.102	5.741	2.533
*p*		0.181	0.052	0.059	0.992	0.001 *	0.059
Difference						5–6	
Working Time in the Profession	<5 years ^1^	29.56 ± 6.68	30.54 ± 6.02	20.2 ± 4.5	20.17 ± 3.24	9.41 ± 1.8	109.88 ± 17.09
	6–10 years ^2^	26.75 ± 7.05	27.75 ± 8.18	16.38 ± 3.46	16.88 ± 1.65	9.5 ± 1.44	97.25 ± 15.58
	11–15 years ^3^	30.08 ± 6.67	29.77 ± 5.62	18.85 ± 3.27	18.77 ± 2.52	8.54 ± 1.29	106 ± 15.55
	16–20 years ^4^	28.6 ± 6.09	30.96 ± 6.14	18.84 ± 4.89	20.12 ± 1.67	8.74 ± 1.19	107.26 ± 12.86
	21–25 years ^5^	26.81 ± 8.03	28.19 ± 5.66	19.25 ± 3.32	19.75 ± 2.51	9.06 ± 1.58	103.06 ± 14.39
	26 years and above ^6^	25.95 ± 8.01	28.45 ± 7.29	20.5 ± 4.07	19.73 ± 2.52	8.39 ± 1.22	103.02 ± 19.42
Test ^a^ (F)		2.946	2.187	4.385	7.322	4.704	3.477
*p*		0.013 *	0.055	0.001 *	0.001 *	0.001 *	0.004 *
Difference		5–6		3–5	2–4	1–3	1–3
Working Hospital	MTRH	28.86 ± 7.1	30.33 ± 6	20.78 ± 3.62	20.17 ± 2.72	9.01 ± 1.6	109.14 ± 16.39
	TÖMC	27.82 ± 7.14	28.92 ± 6.71	17.88 ± 4.44	19 ± 2.75	9.04 ± 1.55	102.66 ± 15.99
Test ^b^ (*t*)		1.735	4.009	42.333	14.869	0.027	13.017
*p*		0.189	0.046 *	0.001 *	0.001 *	0.870	0.001 *

sd: standard deviation. Test ^a^: F value of ANOVA. Test ^b^: *t* value of independent *t*-test. * *p* < 0.05. There is a statistically significant difference between the groups. ^1^: 1th group variable, ^2^: 2th group variable, ^3^: 3th group variable, ^4^: 4th group variable, ^5^: 5th group variable, ^6^: 6th group variable, ^7^: 7th group variable.

**Table 4 healthcare-12-00651-t004:** Comparison of the Work-related Quality of Life scale scores by working unit, working years, working years in the unit, and hospital type.

Variables	Groups	Job and Career Satisfaction	General Well-Being	Control at Work	Working Conditions	Stress at Work	Home-Work Interface	WRQoL Total
		Mean ± sd	Mean ± sd	Mean ± sd	Mean ± sd	Mean ± sd	Mean ± sd	Mean ± sd
Age Group	20–25 ^1^	20.88 ± 4.84	18.64 ± 4.12	9.04 ± 3.33	9.96 ± 3.1	5.52 ± 2.19	10.72 ± 2.74	74.76 ± 17.72
	26–30 ^2^	16.52 ± 3.72	15.57 ± 3.6	5.43 ± 2.19	6.62 ± 2.34	7.05 ± 1.98	7.14 ± 3.07	58.33 ± 10.6
	31–40 ^3^	17 ± 4.35	17.18 ± 3.08	7.36 ± 3.11	7.41 ± 2.31	5.91 ± 1.52	8.82 ± 2.97	63.68 ± 12.43
	41–46 ^4^	19.3 ± 4.67	17.97 ± 3.49	7.7 ± 2.55	8.73 ± 3.01	4.83 ± 2.43	8.86 ± 3.21	67.39 ± 13.45
	47 and above ^5^	18.1 ± 5.84	18 ± 3.48	7.7 ± 3.91	7.7 ± 3.62	5.55 ± 2.08	9.3 ± 3.01	66.35 ± 16.1
Test ^a^ (F)		9.733	7.150	12.018	13.649	9.908	12.481	11.930
*p*		0.001 *	0.001 *	0.001 *	0.001 *	0.001 *	0.001 *	0.001 *
Difference		1–2, 5, 2–4, 3–5	1–2, 4, 2–5, 3–4	1–3, 4, 5, 2–5, 3–5	1–5, 3–4, 4–5	1–2, 4, 3–5, 4–5	1–2, 2–4, 3–4, 4–5	1–2, 3, 4, 3–4, 5, 4–5
Gender	Female	18.77 ± 5.05	17.72 ± 3.76	7.64 ± 3.29	8.17 ± 3.18	5.75 ± 2.36	9.07 ± 3.34	67.13 ± 15.54
	Male	17.5 ± 4.64	16.89 ± 3.53	7.11 ± 3.21	8.11 ± 3	5.79 ± 1.51	8.89 ± 2.77	64.29 ± 14.6
Test ^b^ (*t*)		4.124	3.124	1.653	0.024	0.013	0.199	2.154
*p*		0.043 *	0.078	0.199	0.878	0.909	0.655	0.143
Education Status	High School	19.83 ± 6.71	18.17 ± 4.99	8.28 ± 3.16	8.89 ± 3.43	5.72 ± 2.72	9.56 ± 3.58	70.44 ± 20.32
	Associate Degree	15.86 ± 4.91	16.86 ± 3.35	6 ± 3.19	6.57 ± 3.04	5.71 ± 2.24	9 ± 2.45	60 ± 13.91
	Bachelor’s Degree	18.5 ± 4.88	17.35 ± 3.72	7.53 ± 3.23	8.25 ± 3.14	5.83 ± 2.17	8.87 ± 3.2	66.33 ± 15.3
	Master’s Degree	18.91 ± 4.19	18.82 ± 2.9	7.82 ± 3.59	8 ± 2.77	5.27 ± 1.84	10 ± 3.35	68.82 ± 12.43
Test ^a^ (F)		2.507	1.945	1.934	2.251	0.651	1.401	1.930
*p*		0.059	0.122	0.124	0.082	0.583	0.243	0.125
Worked Unit	Emergency Service ^1^	21.03 ± 4.58	19.23 ± 3.31	8.8 ± 3.63	9.3 ± 3.18	5.8 ± 2.13	10.93 ± 2.96	75.1 ± 15.46
	Operating Room ^2^	18.46 ± 2.93	19.54 ± 2.76	8 ± 4.12	8.85 ± 3.63	5.31 ± 1.25	9 ± 2.2	69.15 ± 13.92
	Intensive Care Unit ^3^	16.46 ± 5	16.54 ± 3.93	5.69 ± 2.43	7.15 ± 3.1	6 ± 1.99	7.23 ± 2.76	59.08 ± 14.48
	Polyclinic Unit ^4^	11.6 ± 3.74	15.8 ± 4.16	4.2 ± 1.66	5.6 ± 2.23	5 ± 2.85	7.4 ± 2.97	49.6 ± 10.36
	Internal Branch Services ^5^	16.77 ± 4.83	15.7 ± 3.48	6.6 ± 2.84	7.17 ± 2.96	6.43 ± 2.35	8.03 ± 3.01	60.7 ± 13.36
	Surgical Branch Services ^6^	19.23 ± 3.6	17.63 ± 3.04	8.03 ± 2.5	8.41 ± 2.43	5.23 ± 1.87	8.83 ± 2.94	67.36 ± 11.01
	Other ^7^	22.36 ± 2.62	20.82 ± 2.96	11.27 ± 1.62	11.36 ± 2.38	4.09 ± 1.38	11.45 ± 2.02	81.36 ± 9.15
Test ^a^ (F)		16.741	11.822	13.222	9.122	3.908	12.882	17.190
*p*		0.001 *	0.001 *	0.001 *	0.001 *	0.001 *	0.001 *	0.001 *
Difference		1–4, 5, 2–4, 5, 3–4, 5–6, 7	1–4, 5, 2–4, 4–6, 7	1–3, 4, 5, 2–4, 3–6, 4–6, 5–6, 7, 6–7	1–4, 3–6, 7	2–3, 6, 3–7, 4–5	1–2, 3, 6, 7, 3–5, 4–5, 5–6	1–3, 4, 5, 6, 7, 2–4, 3–6, 7, 4–6, 7, 5–6, 7, 6–7
Working Time in Unit	<5 years ^1^	18.83 ± 5.1	17.55 ± 3.85	7.63 ± 3.37	8.43 ± 3.12	6.08 ± 2.18	9.09 ± 3.35	67.62 ± 16.11
	5–9 years ^2^	17.22 ± 5.48	17.39 ± 3.61	7.22 ± 3.51	7.17 ± 3.25	5.74 ± 1.84	8.78 ± 3.14	63.52 ± 15.85
	10–14 years ^3^	18.46 ± 3.69	17.22 ± 2.98	7.65 ± 2.6	8.3 ± 2.9	4.7 ± 2.07	9.32 ± 2.5	65.65 ± 11.46
	15–19 years ^4^	17.4 ± 4.01	18 ± 4.29	6 ± 2.85	7.8 ± 2.57	6 ± 2.45	8.4 ± 3.31	63.6 ± 11.94
	20–25 years ^5^	22.5 ± 1.64	20 ± 4.38	9.5 ± 0.55	11.5 ± 2.74	2.5 ± 0.55	12 ± 1.1	78 ± 10.95
	26–29 years ^6^	18.5 ± 0.55	15.5 ± 1.64	7.5 ± 2.74	7 ± 1.1	5 ± 2.19	6.5 ± 0.55	60 ± 1.1
Test ^a^ (F)		2.041	1.007	1.262	3.361	5.885	2.097	1.766
*p*		0.073	0.413	0.280	0.006 *	0.001 *	0.066	0.119
Difference					5–6	5–6		
Working Time in the Profession	<5 years ^1^	19.54 ± 4.89	17.49 ± 4.1	7.93 ± 3.25	8.9 ± 3.2	5.93 ± 2.15	9.41 ± 3.33	69.2 ± 17.29
	6–10 years ^2^	16 ± 3.35	15.5 ± 3.06	5.75 ± 3.1	6 ± 1.77	7.38 ± 2.04	7.38 ± 3.19	58 ± 8.55
	11–15 years ^3^	17.31 ± 5.83	17.85 ± 3.7	7.69 ± 3.67	8.08 ± 3.31	5.38 ± 1.66	9.15 ± 3.46	65.46 ± 16.79
	16–20 years ^4^	18 ± 3.14	18.4 ± 2.56	7.14 ± 2.57	7.8 ± 2.04	5.94 ± 2.2	8.1 ± 2.83	65.38 ± 8.96
	21–25 years ^5^	17.69 ± 5.57	17.25 ± 4.14	7.13 ± 3.25	8.13 ± 3.71	5.44 ± 2.2	9.25 ± 3	64.88 ± 16.57
	26 years and above ^6^	19.07 ± 5.43	17.64 ± 3.26	7.93 ± 3.55	7.73 ± 3.14	4.91 ± 2.15	9.55 ± 2.82	66.82 ± 13.87
Test ^a^ (F)		3.312	2.135	2.257	4.166	4.948	2.851	2.490
*p*		0.006 *	0.061	0.049 *	0.001 *	0.001 *	0.016 *	0.031 *
Difference		5–6		1–2	3–4	2–3	1–2, 2–6	1–2
Working Hospital	MTRH	19.26 ± 5.3	18.43 ± 3.63	7.81 ± 3.51	8.33 ± 3.44	5.38 ± 2.24	10.18 ± 2.89	69.38 ± 16.18
	TÖMC	17.48 ± 4.37	16.42 ± 3.53	7.14 ± 2.94	7.94 ± 2.72	6.22 ± 2	7.66 ± 3.01	62.86 ± 13.47
Test ^b^ (*t*)		10.736	25.510	3.421	1.273	12.730	59.582	15.383
*p*		0.001 *	0.001 *	0.065	0.260	0.001 *	0.001 *	0.001 *

sd: standard deviation. Test ^a^: F value of ANOVA. Test ^b^: *t* value of independent *t*-test. * *p* < 0.05. There is a statistically significant difference between the groups. ^1^: 1th group variable, ^2^: 2th group variable, ^3^: 3th group variable, ^4^: 4th group variable, ^5^: 5th group variable, ^6^: 6th group variable, ^7^: 7th group variable.

**Table 5 healthcare-12-00651-t005:** Coefficients of the model of the relationship between individual workload and quality of work life.

Dependent Variable	Independent Variable	β_1_	*p*	R^2^
Work-related Quality of Life	VSS	0.98	0.001 *	0.96

β_1_: Standardized regression coefficients, * *p* < 0.05; There is a statistically significant relationship between the scales.

## Data Availability

The data are kept confidential within the scope of the personal data protection law. However, upon reasonable request, the data can be accessed by contacting the corresponding author.

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
