# Peer review of "The Effects of Workload Excess on Quality of Work Life in Third-Level Healthcare Workers: A Structural Equation Modeling Perspective"

_healthcare, 2024, doi:10.3390/healthcare12060651_

Round 1
Reviewer 1 Report
Comments and Suggestions for Authors
Abstract and appropriate keywords. Eventually it will not be necessary to include statistical results in the abstract…
The introduction should present results from similar investigations to better understand the framework and objectives of the study carried out. I propose that the revision of this text includes data relating to investigations carried out. The introduction section should be expanded, it is very brief. There is a final paragraph that summarizes the opinions and points of view presented. This paragraph gives you an idea of the opinion of the author of this article proposal.
Very good: explanation of the validation of the survey!
The choices of statistical techniques seem to be adequate for the objectives.
The processing of data through statistical analysis seems to be well designed, so I have no comments.
The presentation of the diagram in Figure 1 helps to interpret the data, good choice!
When processing the data, support is provided associated with previous investigations, which strengthens the data analysis. It is these references to similar investigations that I proposed to be identified in the Introduction section.
The conclusions are very synthetic... after all, the conclusions must be in line with the expectations that the authors had before carrying out the research. They seem very obvious and that would be what common sense would predict. What would be good was to read those more detailed, more detailed aspects, like: it could be more obvious…
The bibliography is varied and most of it is recent, which adds credibility and value to the proposal presented.
Author Response
Dear Reviewer,
- Additions were made to the introduction section and the last paragraph regarding the importance of the research was added.
- The conclusion section has been revised
- English spelling has been checked by experts and errors have been corrected.
- The background of the major adjustments made is highlighted in yellow. However, the hover color was not used for minor corrections during language editing.
Thank you very much for your contributions.
Reviewer 2 Report
Comments and Suggestions for Authors
Although the study looks interesting as the topic is related to the public health issue but the authors have used bulk of statistics and methodologies to prove their point. here i have few points for improvement and be acceptable at statistical level.
1.The authors have correctly used numbers and percentages for the qualitative variables.
2. Why did the authors have used mean and sd for scale questionnaires. Is it acceptable to use mean and Sd for qualitative questions. How will you justify the interpretations of 28.38 mean for manager support. Is mean manager support around 28.38. it has no meanings in real life.
3. All the scale questions have mean and SD. if it seems logical in real life then their interpretations should be added in description of results.
4. Why the authors used structural equation modelling. what was the logic behind. Can logistic or multinomial be applied?
5. The authors have used ANOVA for comparing means. but why did they use ANOVA for this purpose. You can use the percentages and compare their proportions by using chi-square test or any other non-parametric test.
6. The study should include the graphical representation of important qualitative variables using bar-charts or multiple bar charts with respect to gender , education level and other demographic variables for better understanding.
7. The authors should revise the complete statistical analysis and consult the expert statistician for the selections of tests and methods.
8. While stating significance do not use statistical significance. only use significant word in the whole study.
Comments on the Quality of English Language
The english should be improved for better understanding.
Author Response
Dear Reviewer,
The document regarding your evaluation is attached.

Reviewer 3 Report
Comments and Suggestions for Authors
1. Introduction must be rewritten, with regards both to the English language (should be revised) and the content. It does not clarify why the study is important and what it can add to the already available literature. As a side note, also a focus on the workload issue in the healthcare sector is missing.
2. What authors means with "IP address checks were carried out for data security"? If participants answered from the hospitals, it is normal that several responses were received from a single IP address.
3. Lines 98-100: please rephrase.
4. Paragraph 2.4 Data collection tools must be rewritten, it does not use scientific English.
5. Results are very difficult to be interpreted. As a general rule, data shown in tables should not be repeated in text. All the text (e.g. in line 185) starting with "There is a statistically significant difference..." should be removed as it is only a repetition of information from tables. Also, table 3 and 4 are too big. Authors should emphasize only the most relevant information, moving the complete analysis to a Supplementary/Appendix.
5. The discussion should not contain results (e.g. as in line 266), but only the interpretation. Also, this section of the manuscript seems more a "comparison of results with previously published studies" rather than a critical interpretation of results within the framework of already published studies.
Comments on the Quality of English Language
The entire manuscript must be revised. Despite the absence of grammar errors, the used language lacks in scientific soundness.
Author Response
Dear reviewer,
- Additions were made to the introduction section and the last paragraph regarding the importance of the research was added.
- Lines 98-100 were rewritten in line with your suggestion.
- The writing of the data collection tools has been revised.
- The discussion section was reviewed upon your suggestion and the location was changed without any revised additions.
- The conclusion section has been revised
- English spelling has been checked by experts and errors have been corrected.
- The background of the major adjustments made is highlighted in yellow. However, the hover color was not used for minor corrections during language editing.
Thank you very much for your contributions.
Reviewer 4 Report
Comments and Suggestions for Authors
The content and structure of the paper are clear and in accordance with the current rules for writing a scientific article
Author Response
Dear reviewer,
Thank you very much for your evaluation, I respect you.
Round 2
Reviewer 2 Report
Comments and Suggestions for Authors
The authors have completely incorporated all the suggestions.
Author Response
Dear Reviewer,
We tried to do as much as we could the points mentioned by the valuable reviewer in the first revision. We were happy to know that the changes we made met the referee's wishes. Thank you very much for your evaluation, I respect you.